# Corrosion Performance of Ti6Al7Nb Alloy in Simulated Body Fluid for Implant Application Characterized Using Macro- and Microelectrochemical Techniques

Andrea Abreu-García [1], Ricardo M. Souto [1,2,*] and Javier Izquierdo [1,2,*]

1 Department of Chemistry, Universidad de La Laguna, P.O. Box 456, 38200 La Laguna, Tenerife, Spain; andrea.abreu.28@ull.edu.es
2 Institute of Material Science and Nanotechnology, Universidad de La Laguna, 38200 La Laguna, Tenerife, Spain
* Correspondence: rsouto@ull.es (R.M.S.); jizquier@ull.edu.es (J.I.)

**Abstract:** In this paper, the applicability of Ti6Al7Nb as a more biocompatible alternative for bone and dental implants than Ti6Al4V and pure titanium in terms of corrosion resistance and electro-chemical inertness is investigated. The chemical inertness and corrosion resistance of the Ti6Al7Nb biomaterial were characterized by a multi-scale electrochemical approach during immersion in simulated physiological environments at 37 °C comparing its behavior to that of c.p. Ti, Ti6Al4V, and stainless steel. The establishment of a passive regime for Ti6Al7Nb results from the formation of a thin layer of metal oxide on the surface of the material which prevents the action of aggressive species in the physiological medium from direct reaction with the bulk of the alloy. Conventional electrochemical methods such as potentiodynamic polarization and electrochemical impedance spectroscopy (EIS) provide quantified information on the surface film resistance and its stability domain that encompasses the potential range experienced in the human body; unfortunately, these methods only provide an average estimate of the exposed surface because they lack spatial resolution. Although local physiological environments of the human body are usually simulated using different artificial physiological solutions, and changes in the electrochemical response of a metallic material are observed in each case, similar corrosion resistances have been obtained for Ti6Al7Nb in Hank's and Ringer's solutions after one week of immersion (with a corrosion resistance of the order of $M\Omega$ cm$^2$). Additionally, scanning electrochemical microscopy (SECM) provides in situ chemical images of reactive metal and passive dielectric surfaces to assess localized corrosion phenomena. In this way, it was observed that Ti6Al7Nb exhibits a high corrosion resistance consistent with a fairly stable passive regime that prevents the electron transfer reactions necessary to sustain the metal dissolution of the bulk biomaterial. Our results support the proposition of this alloy as an efficient alternative to Ti6Al4V for biomaterial applications.

**Keywords:** implant biocompatibility; Ti6Al7Nb; corrosion resistance; passivity; electrochemical activity; simulated physiological environment

## 1. Introduction

Titanium and titanium-based alloys constitute the main type of biomaterials used for the replacement of lost or diseased bone and dental parts, due mainly to their low density (4.7 g/cm$^3$) compared to other metallic biomaterials such as stainless steel (7.9 g/cm$^3$) or the Co–Cr–Mo alloy (8.3 g/cm$^3$) and their mechanical properties which are closer to those of bone (e.g., Young's modulus less than 110 GPa) [1–6]. These are materials of great chemical inertness, low toxicological effects, and high resistance to corrosion, which render them highly biocompatible [7–9]. Their chemical inertness and corrosion resistance derive from the development of a somewhat stable passivity regime due to the spontaneous formation of a compact oxide layer, generally TiO$_2$, both in air and in

aqueous solution, thus protecting the bulk material from attack by oxidants present in the physiological environment [7,10]. Unfortunately, this passivating oxide layer undergoes localized decomposition in the ubiquitous presence of chloride ions [11–13], as is the case in physiological and marine environments [14]. The combined effect of these aggressive ions and electrical polarization enhances passive-layer thinning and dissolution rates and triggers hydrogen evolution in these materials [15–20]. Therefore, electrochemical methods are employed to investigate the stability and modification of the protective oxide layers in simulated physiological conditions [21–24], as well as the protectiveness and effects of surface-modification procedures in terms of composition and thickness [25–28].

Although Ti6Al4V is the most widely used biomaterial [29], there is growing concern due to carcinogenic reports related to the release of vanadium [30–32]. More controversial is the eventual involvement of aluminum in some neurological diseases [32,33], whereas alloying aluminum benefits microstructure control which is of interest for implantation. As a result, Ti6Al7Nb has been proposed as a more biocompatible alternative to Ti6Al4V [34]. In fact, Ti6Al4V and Ti6Al7Nb present similar microstructural characteristics consistent with a Widmansttaten structure formed by coarse β-phase grains containing a lamellar α-phase [35]. On the other hand, the substitution of vanadium by niobium cations in the Ti6Al4V alloy has been claimed to improve the passivation properties of the surface, eliminating the stoichiometric defects (anionic vacancies) present in the titanium-dioxide layer [36].

The corrosion of metallic implant materials in a physiological environment is due to electrochemical processes [9,24,34]; that is, distributed local microcells consisting of anodes (sites where metal oxidation occurs) and cathodes (sites where certain chemical species are reduced) form because the material is in direct contact with an aqueous electrolytic fluid phase. The corrosion rate is affected by local chemical heterogeneities in the material which give rise to a galvanic coupling [10]. Although cathodic and anodic sites may sometimes occur close to each other on the metal surface, the two reactions cannot occur simultaneously at the same location. Since these processes are electrochemical in nature, they can be best studied using electrochemical techniques [23,25,35,37,38]. Unfortunately, the knowledge gathered until now on the corrosion resistance and stability of the passive regime of Ti6Al7Nb has been solely obtained using classical electrochemical techniques, including potentiodynamic polarization and electrochemical impedance spectroscopy (EIS) [26,38]. Given that such techniques are based on signals resulting from averaging the response of all surface sites, they lack spatial resolution and are thus unable to resolve the contributions of the various microstructural components of the alloy. Spatially resolved measurements of the electrochemical activity on the complex microstructures of titanium-based alloys have become possible by using scanning electrochemical microscopy (SECM) in amperometric operation [39], a technique that scans a polarizable microelectrode (ME) close to the surface of the alloy under study. In this way, the kinetics of electron transfer rate reactions and the dielectric characteristics of passivated nitinol [18,40,41] and Ti6Al4V [42,43] have been determined as a function of the applied polarization and compared to the properties of the passive film on pure titanium [19,44]. However, to the best of our knowledge, the characterization of surface-reactivity distribution by such a procedure remains unexplored for Ti6Al7Nb until now.

In this paper, we report a multiscale electrochemical characterization of Ti6Al7Nb in artificial physiological solution at 37 °C, comparing its behavior with other commercial titanium-based biomaterials and various stainless steels. Electrochemical characterization was performed using potentiodynamic polarization, electrochemical impedance spectroscopy, and scanning electrochemical microscopy.

## 2. Materials and Methods

### 2.1. Materials

Rods of as-cast Ti6Al7Nb were produced by the Rare and Non-Ferrous Metals Institute (Bucharest, Romania), while commercial rods of c.p. Ti (grade 2) and Ti6Al4V (grade 5)

were purchased from Nobel Biocare (Kloten, Switzerland). For the sake of comparison, plates of stainless steel of biomaterial grade 316L (Fe18Cr10Ni3Mo) and nonbiomaterial grade 304 (Fe18Cr10Ni) were supplied by Goodfellow (Cambridge, UK).

The samples employed for electrochemical characterization were prepared by embedding the metallic material in an inert epoxy resin (Epofix, Struers, Ballerup, Germany) in order to control the surface area exposed to the test solution, as well as to regenerate a reproducible clean surface through surface abrasion using a sequence of SiC abrasive paper ranging from 800 to 4000 in grit size. The samples were degreased and ultrasonically cleaned in ethanol and then dried under air.

The electrochemical behavior of the different materials was tested in two simulated physiological media types, namely Ringer's and Hank's solutions, thermostatted at 37 °C. Ringer's solution was prepared with the composition of 8.50 g/L NaCl, 0.400 g/L KCl, and 0.340 g/L $CaCl_2 \cdot H_2O$, and the composition of Hank's solution was as follows: 8.00 g/L NaCl, 0.400 g/L KCl, 0.185 g/L $CaCl_2 \cdot H_2O$, 0.200 g/L $MgSO_4 \cdot 7H_2O$, 0.0600 g/L $Na_2HPO_4$, 0.0460 g/L $KH_2PO_4$, and 1.00 g/L glucose. These solutions have been chosen in order to match the average mineral composition of the physiological fluid in the human body, but only the latter exhibits pH-buffering characteristics. The redox mediator employed for electrochemical imaging using SECM was ferrocene-methanol (Sigma Aldrich, St. Louis, MO, USA), which was added to the simulated physiological solution at a concentration of 0.5 mM. All solutions were prepared using analytical grade reactants and deionized Milli-Q®-grade water.

### 2.2. Methods

Classical electrochemical testing was performed in a three-electrode corrosion cell using PARSTAT® (model 2263 potentiostat/galvanostat, PowerSUITE® software) from Princeton Applied Research (Ametek, Berwyn, PA, USA). A saturated calomel electrode (SCE) and a platinum grid were used as reference and auxiliary electrodes, respectively. All potential values herein are referenced to SCE unless otherwise indicated. The experimental testing sequence for each sample consisted of recording the open circuit potential (OCP) of the material in the test solution for 1 h, followed by recording a series of electrochemical impedance spectra (EIS) at various elapsed times for up to 1 week. The amplitude of the sinusoidal AC signal was 10 mV around the OCP at frequencies between $10^5$ and $10^{-2}$ Hz. Finally, a cyclic potentiodynamic curve was recorded in the potential range between −0.25 V vs. OCP and +1.10 V at a scan rate of 1 mV/s. The initial scan direction was towards more positive potential values.

Microelectrochemical testing was performed using a SECM instrument built by Sensolytics (Bochum, Germany) containing an Autolab bipotentiostat (Metrohm Autolab, Utrecht, Netherlands), as sketched in Figure 1. Amperometric operation in the feedback mode was performed with a Pt microelectrode (ME) tip of 25 μm in diameter connected to the WE#1 input of the bipotentiostat using ferrocene-methanol as a redox mediator [45]. The geometrical factors of the ME were characterized by recording a cyclic voltammogram for the oxidation of the redox mediator in Ringer's solution modified by the addition of 0.5 mM ferrocene-methanol, as shown in Figure 2. The reference was an Ag/AgCl/KCl (sat.) electrode, and a Pt wire served as the auxiliary electrode. In SECM operation under the feedback mode, the variation in the measured stationary diffusion current, $i_{lim} = i_{T,\infty}$, at the surface of the ME while it is scanned near the surface of the substrate provides spatially resolved information on the local electrochemical activity of the studied metal. Normalized current values, $I$, were determined by dividing the measured current values, $i_T$, by $i_{lim}$. The tip potential for the oxidation of the redox mediator was set at +0.50 V, and the scan rate for 2D mapping was 15 μm/s, and the step size was 15 μm. The operating tip-substrate distance was established by recording a Z-approach curve on a location over a resin area close to the metal sample until the current measured at the tip dropped to 50% of the $i_{T,\infty}$ value recorded in the bulk of the electrolyte (see Figure 2); that is, approximately

15 μm, which is then the spatial resolution of the 2D maps. In selected experiments, the polarization of the metal sample was performed using the WE#2 input of the bipotentiostat.

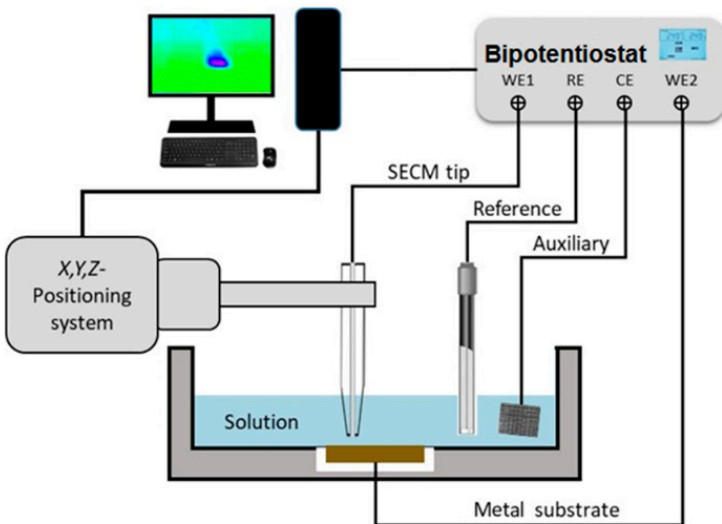

**Figure 1.** Sketch depicting the SECM setup and electrode connections for amperometric operation (adapted from ref. [45]).

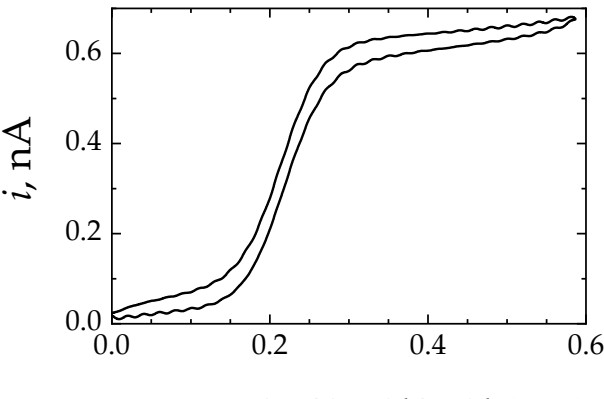

**Figure 2.** Typical cyclic voltammetry curve recorded at the Pt ME in Ringer's solution + 0.5 mM ferrocene-methanol. ME diameter, 25 μm; scan rate: 50 mV/s.

## 3. Results

Two simulated physiological solutions were used in this work for the multiscale electrochemical characterization of Ti6Al7Nb at 37 °C, namely Ringer's and Hank's solutions. Subsequently, Ringer's solution was chosen for the comparison of this alloy with samples of two other alloy groups of materials due to the simpler composition of this medium. The selection of the two groups of alloys, namely titanium-based and stainless steel (SS) samples, was motivated by two aims: firstly, to compare the performance of Ti6Al7Nb with the most widely employed Ti-based biomaterials (c.p. Ti and Ti6Al4V); secondly, stainless steels were selected mainly to compare the electrochemical characteristics of other passive systems exhibiting surface-film breakdown within the potential range recorded under infection conditions in the human body [46]. In the latter case, the selected stainless steels included biomaterial-quality 316L SS, whereas 304 SS is not adequate for implantation. The electrochemical results are described first on the basis of conventional electrochemical testing using open-circuit electrochemical impedance spectroscopy and potentiodynamic polarization measurements, followed by spatially resolved surface characterization using scanning electrochemical microscopy.

### 3.1. Macroelectrochemical Characterization

Immediately after preparation of the Ti6Al7Nb samples, their open-circuit potential (OCP) was recorded as a function of time in Ringer's and Hank's solutions. As shown in Figure 3, the OCP values in Hank's solution stabilize faster at a less noble stationary potential (ca. −0.30 V vs. SCE), suggesting that in this solution, the corrosion resistance is lower for this alloy. In this case, the presence of phosphate and carbonate ions can facilitate the formation of a passive layer more quickly, although with less protective characteristics. In contrast, after a transient state of electrochemical reactivity during approximately 250 s after immersion in Ringer's solution, associated with the formation and subsequent compaction of the passivating oxide layer, the potential evolves towards more positive values during the rest of the measurement. This increase in potential reflects a less active electrochemical behavior of the surface due to the development of a thicker passive layer until reaching a dynamic equilibrium condition.

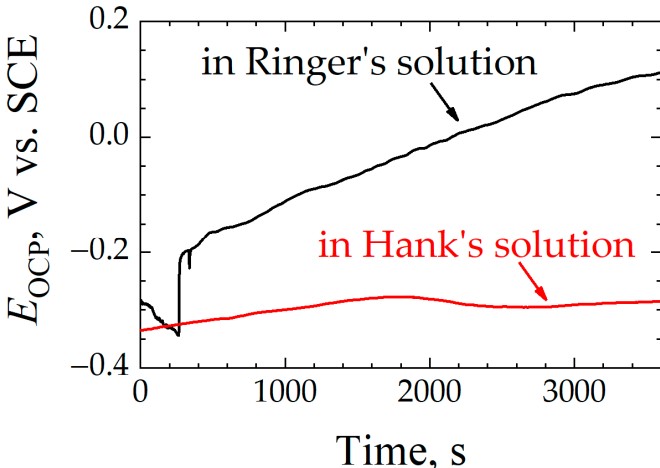

**Figure 3.** Open-circuit potential ($E_{OCP}$) versus time for Ti6Al7Nb immersed in a simulated physiological solution at 37 °C. Test solution: (black line) Ringer's, and (red line) Hank's.

The evolution of OCP values towards more positive values due to the formation of a more protective oxide layer on the Ti6Al7Nb alloy after immersion in Ringer's solution was also observed for the other alloys studied in this work. In Figure 4, the potential rises to a stable value for most materials, albeit without showing the initial drop to more negative potentials displayed by Ti6Al7Nb, which was followed by a less stationary system for the rest of the experiment. Both c.p. Ti and 316L SS samples reached a constant potential value within short times. For 304 SS, although it managed to quickly reach a stable potential value, it underwent, at certain moments in time, abrupt drops in potential that subsequently returned to the previous values—a sign of rupture of the passive layer and momentary loss of its protective character until it is protected again. This is evidence of the dynamic nature of the oxide layer developed on 304 SS that undergoes transient film breakdown and metastable pitting in this chloride-containing solution [47]. A similar process occurs with the Ti6Al4V sample, although to a lesser degree; although the oxide layer formed on Ti6Al4V is of a more compact and protective nature than those formed on 304 SS and similar stainless steels [23,35], it also experiences some degree of passivity breakdown in Ringer's solution at 37 °C, and yet it quickly repairs without showing signs of localized corrosion propagation [13]. In general, for passivated metallic materials, the higher the OCP of the metal, the more corrosion resistance they will exhibit. Therefore, the Ti6Al7Nb, c.p. Ti, and 316L SS samples that show the highest OCP values in Figure 4 after 1 h immersion can be regarded as the most resistant to corrosion in this environment. In contrast, 304 SS exhibits the lowest potential values, revealing a more electrochemically active surface.

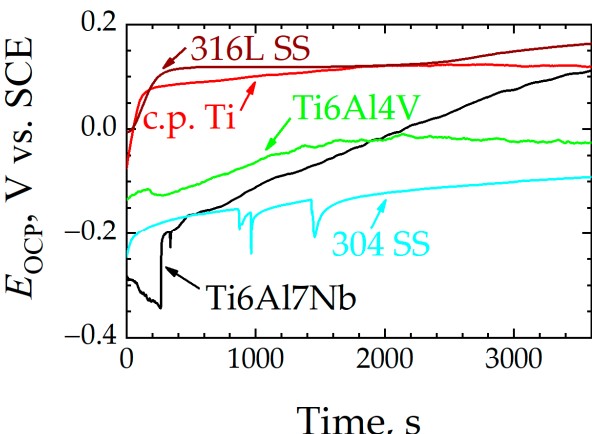

**Figure 4.** Open-circuit potential ($E_{OCP}$) versus time in Ringer's solution at 37 °C for: (black line) Ti6Al7Nb, (green line) Ti6Al4V, (red line) c.p. Ti, (dark red line) 316L SS, and (blue line) 304 SS.

Next, we studied the evolution of the passive layer formed on the biomaterial sample in regard to the electrochemical characteristics and eventual aging along immersion in simulated body fluid for up to 1 week using electrochemical impedance spectroscopy (EIS). Impedance spectra in Bode form obtained at the open-circuit potential for Ti6Al7Nb in Ringer's and Hank's solutions are given in Figure 5. Regardless of the duration of exposure and the composition of the testing environment, the impedance spectra are dominated by phase angles greater than 45° in a wide frequency range, a feature typically related to a capacitive behavior with a high resistance towards electron transfer reactions on the surface. This capacitive behavior is more evident in the case of Ringer's solution, where values close to 80° are observed in a wide frequency range. This indicates a broad capacitive response of the material, which is attributed to the presence of a protective passive layer. This capacitive behavior improves even more after 1 week of exposure to the medium, as it extends up to higher frequency ranges (ca. $10^3$ Hz). However, when Ti6Al7Nb is exposed to Hank's solution, this capacitive behavior worsens as the exposure time elapses, as both the phase angle decreases (maximum of only 65°) and the frequency range in which it remains constant becomes narrower. Moreover, the appearance of two maxima is better resolved, a feature showing that the passive layer becomes more porous as the exposure time progresses. On the other hand, it is observed that in Ringer's solution, the impedance modulus at the low frequency limit (effectively determined at 10 mHz) increases as the exposure time increases, reaching values of $4.9 \times 10^5 \ \Omega \ cm^2$ at the end of the experiment. In contrast, when this material is exposed to Hank's solution, only small changes in the low frequency of the impedance modulus can be observed from inspection of Figure 5C, in spite of the shape of the Bode-phase graphs changing significantly in Figure 5D. Thus, the impedance modulus shows a maximum of $3.0 \times 10^5 \ \Omega \ cm^2$ after one day of exposure and then slowly decreases to values close to those determined at the beginning of the immersion (ca. $2.6 \times 10^5 \ \Omega \ cm^2$). Although there are certain differences, the electrochemical behavior of Ti6Al7Nb in both electrolytic media at their OCP is characterized by the presence of a passive oxide film with dielectric characteristics. These EIS observations are consistent with the extrapolation of the OCP observations over extended time periods, reflecting a quick formation of a stable yet poorly protective (eventually porous) passive layer in carbonate and phosphate-containing Hank's medium, whereas Ringer's solution promotes the progressive development of a protective film with a greater ability to isolate the metal from the environment.

A more detailed inspection of the spectra shows that the electrochemical behavior of the alloy is better described by identifying two time constants, an electrochemical behavior typically found for pure titanium [26,40] and some of its alloys [16,21,22,24,48] (including Ti6Al7Nb at ambient temperature [35,40]). This is regarded to result from the development of a bilayer oxide film on the surface that is composed of a relatively thin and very compact

inner oxide film and a thicker and somewhat porous outer layer [24]. The latter only partially seals the surface and contains pores that can be filled by the aggressive electrolyte, leading to the formation of ionic pathways.

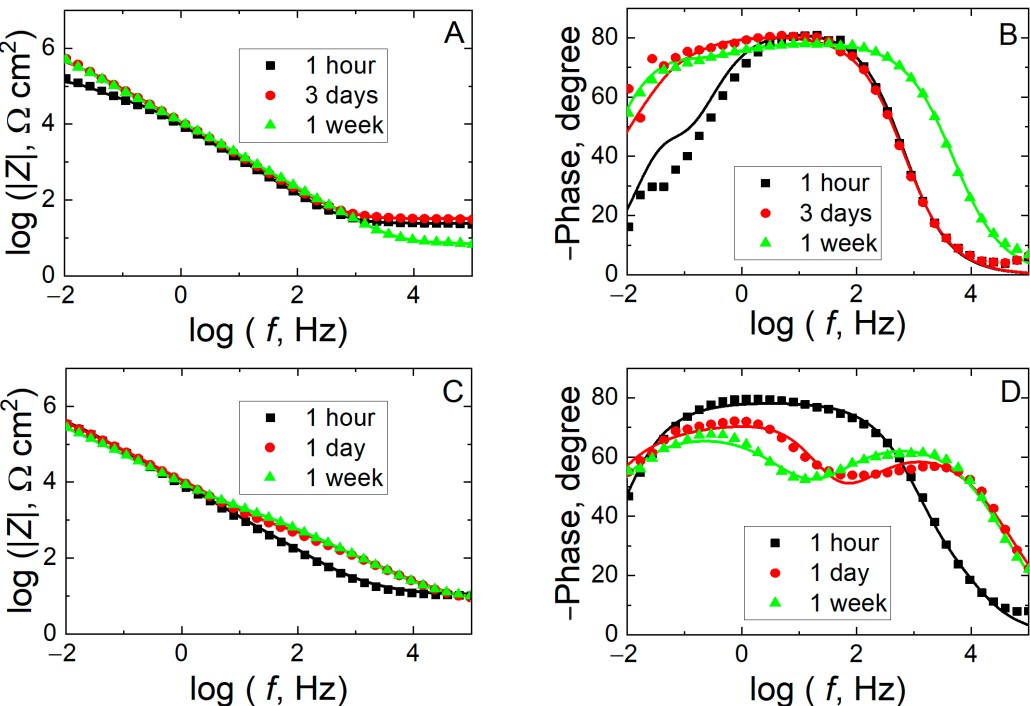

**Figure 5.** Measured (discrete points) and fitted (solid lines) impedance spectra for Ti6Al7Nb alloy recorded at the corresponding open-circuit potential during exposure to a simulated physiological solution at 37 °C for various times as indicated. Test solution: (**A,B**) Ringer's and (**C,D**) Hank's. (**A,C**) Bode-amplitude and (**B,D**) Bode-phase diagrams.

Such physicochemical behavior can be described using the equivalent circuit shown in Figure 6, and its electrical parameters can be employed to model and quantify the system [24]. At the highest frequencies, the system behaves purely resistively, and $R_\Omega$ describes the uncompensated ohmic resistance of the simulated fluid solution. The subsequent increase in both the impedance amplitude and the phase angle in the Bode plots with decreasing frequency reveals the impedance characteristics arising from the penetration of the electrolyte into the pores of the outer oxide layer, which is described by the resistance $R_p$ and the capacitance $Q_p$ of the outer film. At lower frequencies, the high resistive behavior of the impedance values is associated with the sealing effect of the inner thin oxide film characterized by $R_b$ and $Q_b$, respectively. Due to surface roughness and chemical heterogeneities, constant phase elements (CPE, $Q$) were used instead of ideal capacitances, $C$ [49]. Capacitance values can be derived from the CPE parameters ($Y_0$, $n$) using [50]:

$$C = (R^{1-n} \, Y_0)^{1/n} \tag{1}$$

This fitting procedure was employed to determine the values of the impedance parameters for Ti6Al7Nb samples immersed in Hank's and Ringer's solutions for various exposure times, as listed in Table 1. When the Ti6Al7Nb alloy is immersed in Ringer's solution, the values of $R_b$ increase considerably during the first day of exposure, increasing from an order of $10^4 \ \Omega \ cm^2$ to almost $10^6 \ \Omega \ cm^2$ and then remaining in this order of magnitude until completing the experimental series. This feature evidences a reinforcement of the passive layer after completing 1 day of exposure and its permanence with time. However, when the material is exposed to Hank's solution, $R_b$ remains somewhat constant throughout the exposure period at values below $10^6 \ \Omega \ cm^2$, although the highest value is

observed after 1 day. This feature may indicate that the passive layer is fairly protective and stable. Nevertheless, $R_p$ significantly differs in both conditions, reaching up to the $10^5$ $\Omega$ cm$^2$ range in Ringer's solution yet never reaching above the k$\Omega$ cm$^2$ range in Hank's solution. This observation suggests that the outer oxide film is indeed a determinant for the material's performance, even behaving as a leaky capacitor ($n_p$ between 0.60 and 0.88) when compared to that formed in Ringer's environment ($n_p$ always approximately 0.9).

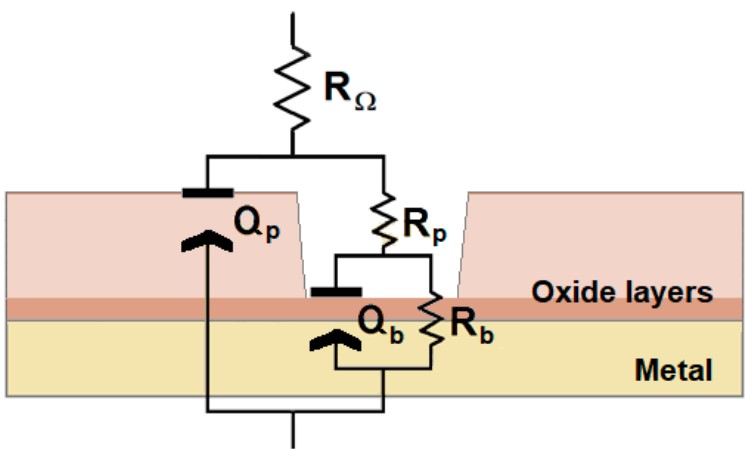

**Figure 6.** Equivalent circuit (EC) used in the generation of simulated data in the electrochemical impedance spectra.

**Table 1.** Impedance parameters of Ti6Al7Nb samples in Ringer's and Hank's solutions at 37 °C.

| Solution | Immersion Time/h | $R_p$/k$\Omega$ cm$^2$ | $10^5$ $Q_p$/ S cm$^{-2}$ s$^n$ | $n_p$ | $R_b$/ k$\Omega$ cm$^2$ | $10^5$ $Q_b$/ S cm$^{-2}$ s$^n$ | $n_b$ |
|---|---|---|---|---|---|---|---|
| | 1 | $63.6 \pm 13.8$ | $1.85 \pm 0.14$ | $0.92 \pm 13.8$ | $94.7 \pm 43.13$ | $8.82 \pm 6.19$ | $1.00 \pm 0.32$ |
| Ringer's | 72 | $38.8 \pm 7.2$ | $1.23 \pm 0.07$ | $0.95 \pm 0.01$ | $835 \pm 59$ | $1.12 \pm 0.16$ | $0.84 \pm 0.12$ |
| | 168 | $197 \pm 120$ | $1.70 \pm 0.06$ | $0.87 \pm 0.01$ | $1057 \pm 143$ | $0.40 \pm 0.14$ | $0.90 \pm 0.13$ |
| | 1 | $0.022 \pm 0.020$ | $1.21 \pm 1.49$ | $0.88 \pm 0.10$ | $721 \pm 20$ | $1.01 \pm 1.48$ | $0.72 \pm 0.02$ |
| Hank's | 24 | $1.42 \pm 0.20$ | $1.51 \pm 0.09$ | $0.73 \pm 0.01$ | $1722 \pm 109$ | $0.60 \pm 0.11$ | $0.92 \pm 0.04$ |
| | 168 | $3.20 \pm 0.65$ | $1.34 \pm 0.15$ | $0.74 \pm 0.01$ | $978 \pm 80$ | $1.36 \pm 0.16$ | $0.78 \pm 0.02$ |

For the sake of comparison, the electrochemical impedance spectroscopy responses of c.p. Ti, Ti6Al4V, 316L SS and 304 SS were recorded in Ringer's solution. From a cursory view of the impedance spectra depicted in Figure 7, the electrochemical response of the stainless steels can be distinguished from those of the titanium-based materials. In fact, whereas a mostly capacitive behavior accompanied by high values of the impedance magnitude values can be observed for c.p. Ti and Ti6Al4V, smaller impedance values and narrower and shallower capacitive loops are observed for 304 SS and 316L SS. Despite these differentiating features, two time constants are still observed in all cases, and the EC of Figure 6 can be employed for the analysis and comparison of the impedance spectra. The values of the impedance parameters for each material are given in Table 2.

The comparison between the different materials was made on the basis of the parameter $R_b$, which accounts for the main barrier to electron transfer (i.e., electrochemical reactivity) for the surface covered by a compact film; the impact on the total resistance of the passive layers is shown in Figure 8 in the form of bar diagrams. In most cases, high values are observed, evidencing the existence of a highly protective passive layer. For the Ti6Al4V sample, after one day of exposure, the values of $R_b$ increase up to an order of $10^7$ $\Omega$ cm$^2$, but after one week of exposure, they abruptly decrease to approximately $10^3$ $\Omega$ cm$^2$, a value that is even smaller than that observed at the beginning of the experiment (approximately $10^5$ $\Omega$ cm$^2$). This fact reveals that although there is some reinforcement of the passive layer upon immersion in the solution during the initial 24 h, the subsequent decrease must be

related to the partial dissolution and increased porosity of the oxide layer. In contrast, in the case of c.p. Ti, $R_b$ gradually increases, indicating that the passive layer becomes more protective over time, stabilizing even after one week. This behavior is similar to that described for the Ti6Al7Nb alloy in Ringer's solution during the initial days, since $R_b$ increases considerably during the first day of exposure, from approximately $10^4$ $\Omega$ cm$^2$ to almost $10^6$ $\Omega$ cm$^2$. Although the individual tendencies of $R_b$ in c.p. Ti and Ti6Al7Nb differ (i.e., maximum values are found after one week and one day for c.p. Ti and Ti6Al7Nb, respectively), the total resistance progressively increases over time for both materials.

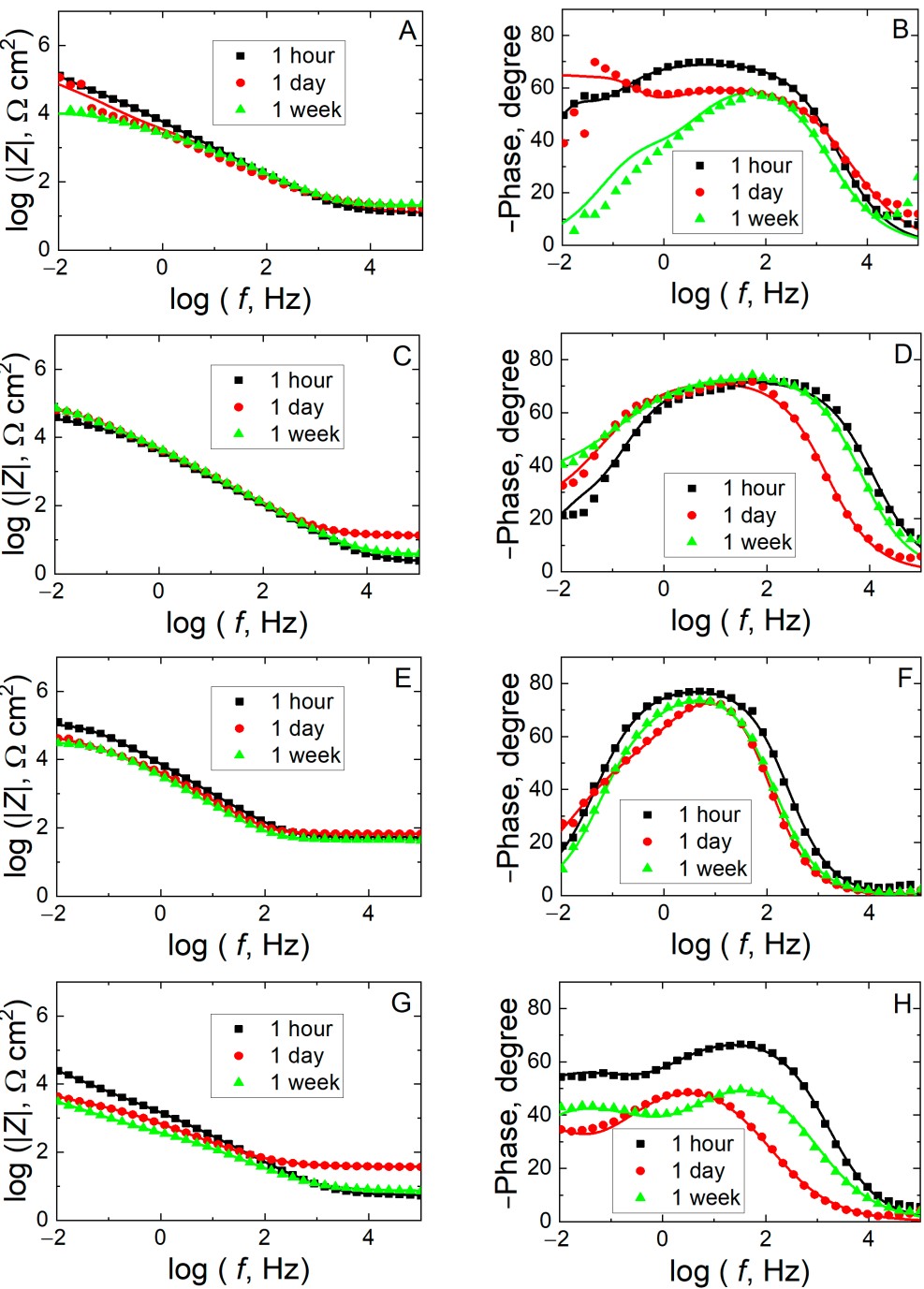

**Figure 7.** Measured (discrete points) and fitted (solid lines) impedance spectra recorded at the corresponding open-circuit potential during exposure to Ringer's solution at 37 °C for various times as indicated. Material: (**A**,**B**) Ti6Al4V, (**C**,**D**) c.p. Ti; (**E**,**F**) 316L SS, and (**G**,**H**) 304 SS. (**A**,**C**,**E**,**G**) Bode-amplitude and (**B**,**D**,**F**,**H**) Bode-phase diagrams.

**Table 2.** Impedance parameters of c.p. Ti, Ti6Al4V, 316L SS, and 304 SS samples in Ringer's solution at 37 °C.

| Sample | Immersion Time/h | $R_p/$ kΩ cm² | $10^5 Q_p/$ S cm⁻² sⁿ | $n_p$ | $R_b/$ kΩ cm² | $10^5 Q_b/$ S cm⁻² sⁿ | $n_b$ |
|---|---|---|---|---|---|---|---|
| c.p. Ti | 1 | 19.6 ± 17.6 | 6.22 ± 3.01 | 0.80 ± 0.06 | 30.2 ± 26.5 | 32.3 ± 69.4 | 0.82 ± 0.41 |
| | 24 | 69.8 ± 4.1 | 4.55 ± 0.22 | 0.81 ± 0.01 | 98.3 ± 13.1 | 75.4 ± 11.8 | 1.00 ± 0.13 |
| | 168 | 17.9 ± 7.54 | 4.16 ± 0.13 | 0.82 ± 0.12 | 2027 ± 3070 | 3.50 ± 0.60 | 0.40 ± 0.05 |
| Ti6Al4V | 1 | 110 ± 20 | 4.00 ± 0.16 | 0.78 ± 0.01 | 222 ± 44 | 6.23 ± 3.43 | 1.00 ± 0.17 |
| | 24 | 10.3 ± 5.2 | 8.14 ± 1.00 | 0.67 ± 0.01 | 1050 ± 210 | 14.8 ± 3.7 | 1.00 ± 0.10 |
| | 168 | 3.48 ± 1.48 | 5.40 ± 0.98 | 0.73 ± 0.02 | 7.49 ± 2.76 | 14.1 ± 4.94 | 0.72 ± 0.22 |
| 316L SS | 1 | 50.7 ± 10.6 | 2.53 ± 0.22 | 0.89 ± 0.01 | 74.2 ± 5.2 | 0.94 ± 0.07 | 0.42 ± 0.09 |
| | 24 | 7.54 ± 1.70 | 3.53 ± 0.16 | 0.92 ± 0.01 | 60.5 ± 7.3 | 6.05 ± 0.56 | 0.58 ± 0.05 |
| | 168 | 6.61 ± 1.16 | 3.51 ± 0.11 | 0.92 ± 0.01 | 72.1 ± 7.1 | 5.93 ± 0.34 | 0.53 ± 0.03 |
| 304 SS | 1 | 3.69 ± 0.67 | 12.8 ± 0.54 | 0.78 ± 0.01 | 258 ± 38.2 | 14.2 ± 0.82 | 0.63 ± 0.02 |
| | 24 | 3.43 ± 0.41 | 40.9 ± 1.8 | 0.64 ± 0.01 | 28.3 ± 9.2 | 164 ± 41 | 0.64 ± 0.07 |
| | 168 | 0.44 ± 0.08 | 42.0 ± 4.3 | 0.67 ± 0.01 | 13.9 ± 1.9 | 113 ± 6.7 | 0.57 ± 0.03 |

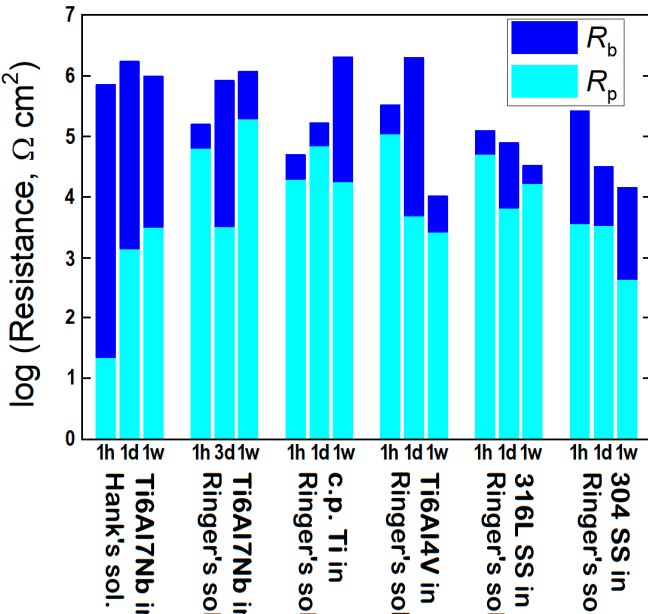

**Figure 8.** Resistance parameters obtained from the analysis of the EIS data using the EC in Figure 6.

In the case of the stainless steels 304 and 316L, they experience a behavior typical of a material with a low stability in prolonged exposure. Thus, for 316L SS, although $R_b$ slightly increases during the first day, it then shows a pronounced decrease after 1 week, which indicates the dissolution of the passive layer and therefore compromises corrosion protection. However, in 304 SS, an abrupt decrease in $R_b$ values mainly occurs during the first day of exposure from approximately $10^5$ Ω cm² to $10^4$ Ω cm², which indicates the incomplete sealing of the passive layer, a trend that is observed to continue at the end of the one-week exposure to the test solution, although with a greater decrease that maintains the value of $R_b$ on the order of $10^4$ Ω cm² during the process of aging.

Finally, upon completing a one-week exposure to the test medium, each of the samples was subjected to potentiodynamic polarization, and the plots obtained for the Ti6Al7Nb alloy in Ringer's and Hank's media are given in Figure 9; Figure 10 shows the plots obtained for the other alloys in Ringer's medium. The electrochemical parameters extracted from the analysis of the potentiodynamic polarization curves shown in Figures 9 and 10 are summarized in Table 3.

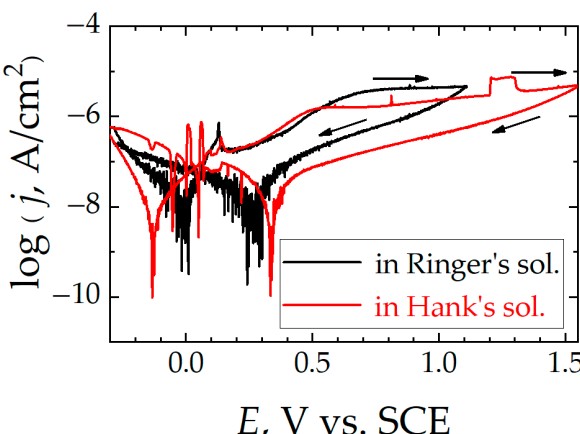

**Figure 9.** Cyclic polarization curves for Ti6Al7Nb after 1 week of immersion in simulated physiological solution at 37 °C. Test solution: (black line) Ringer's, and (red line) Hank's. Scan rate: 1 mV/s.

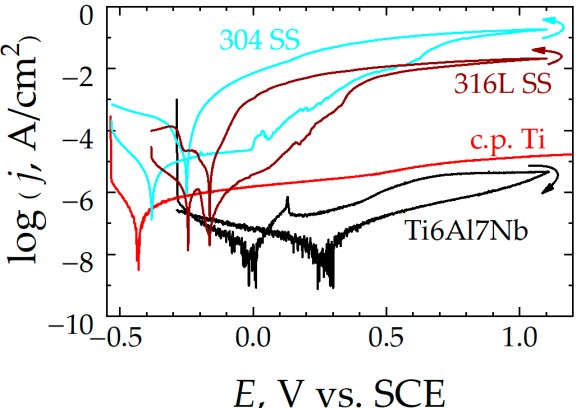

**Figure 10.** Cyclic polarization curves recorded after 1 week of immersion in Ringer's solution at 37 °C for: (red line) c.p. Ti, (black line) Ti6Al7Nb, (dark red line) 316L SS, and (blue line) 304 SS. Scan rate: 1 mV/s. Arrows indicate the sweep direction for potential.

**Table 3.** Electrochemical parameters extracted from the potentiodynamic polarization curves shown in Figures 9 and 10 recorded after 1 week immersion in Ringer's solution at 37 °C for Ti6Al7Nb, Ti6Al4V, c.p. Ti, 316L SS, and 304 SS samples.

| Sample and Solution | $E_{cor}$, V vs. SCE | $j_{cor}$, A/cm$^2$ | $j_{pas}$, A/cm$^2$ | $E_{rep}$, V vs. SCE |
|---|---|---|---|---|
| Ti6Al7Nb in Hank's | −0.132 | $1.25 \times 10^{-8}$ | $1.58 \times 10^{-6}$ | − |
| Ti6Al7Nb in Ringer's | +0.090 | $1.12 \times 10^{-7}$ | $3.16 \times 10^{-6}$ | − |
| Ti6Al4V in Ringer's | −0.183 | $9.47 \times 10^{-8}$ | $5.01 \times 10^{-7}$ | − |
| c.p. Ti in Ringer's | −0.498 | $7.52 \times 10^{-7}$ | $1.60 \times 10^{-6}$ | − |
| 316L SS in Ringer's | −0.185 | $2.78 \times 10^{-7}$ | $4.07 \times 10^{-6}$ | +0.654 |
| 304 SS in Ringer's | −0.354 | $4.18 \times 10^{-6}$ | $2.04 \times 10^{-5}$ | +0.550 |

In general, it is possible to observe four domains in the potentiodynamic curves: first, a current response corresponding to a cathodic behavior due to the reduction of the dissolved oxygen present, since the tests are carried out in the naturally aerated solution. As more positive potentials are applied, it is possible to distinguish a cathodic–anodic transition (that is, the current density changes from a negative to a positive sign), where the magnitude of the faradaic current decreases until it reaches a minimum at the curve at a potential known as the corrosion potential ($E_{cor}$). At this potential, the potentiostat does not impose any overvoltage on the system; thus, the material is capable of behaving

as an electrochemical cell by itself, where anodic processes that involve the oxidation of the metal and cathodic processes that involve the reduction of oxygen occur simultaneously on the exposed surface. In this way, both half-reactions compensate for each other, since the electrons released during metal oxidation are used to reduce oxygen on the metal surface, leading to a net current close to zero. Therefore, the exchange current due to these anodic and cathodic processes, $j_{cor}$, can only be obtained from interpolations on the graph.

In the third region of these curves, as more positive potentials are applied to the samples, they experience anodic behavior; that is, the material oxidizes, increasing the Faradaic current and generating corrosion products capable of creating a passive layer on the surface of the alloy that acts as a dielectric barrier beyond a certain point. This gives rise to a platform where the current density values remain constant in this passivity regime ($j_{pas}$), even though the polarization increases, thus exhibiting resistance to corrosion. However, such a passive state with a stationary current regime is not found for the two stainless steels, thus they are not completely passivated. In addition, as the potential increases, activation events corresponding to metastable pitting and subsequent repassivation can be observed on their polarization curves. In contrast, the best behavior in terms of passivation is observed with the Ti6Al7Nb alloy, since it attains the lowest $j_{pas}$ values both in Ringer's and in Hank's media, and these values remain nearly constant throughout the polarization procedure.

The stability of the stationary passive regime attained by Ti6Al7Nb was further checked by allowing the potential excursion to extend to even higher potential values for the sample immersed in Hank's solution. In this way, the eventual occurrence of transient passive regime breakdown could be monitored. Indeed, the occurrence of a transient breakdown of the passive layer occurred at approximately +1.20 V, as indicated by an event of increased current that subsequently decayed back to the stationary values as the surface became passivated again within a short time and then remained passive up to +1.60 V. Although these anodic potential values are more positive than the highest value recorded in the human body under infection conditions to date (ca. +0.71 V vs. SCE) [42], and the surface of the alloy may be regarded to remain passive, this observation confirms the dynamic nature and transient breaking and rebuilding of the oxide layer in a simulated physiological solution at body temperature that was previously found for the Ti6Al4V alloy [12] and even c.p. Ti [13]. This feature must be taken into account for adequate biocompatibility assessment because transient breakdown of the passive layer is due to the occurrence of metastable pitting [47], and it may account for a certain amount of metal release despite most of the material remaining passive [10].

Potential reversal allows knowledge of the regeneration capacity of the oxide layer formed on the metal surface. Although hysteresis loops are recorded in all cases, a major difference is observed between the titanium-based materials and the stainless steels. Namely, positive hysteresis cycles are found for the latter; that is, the anodic current density values are higher in the negative-direction scan than they were measured to be during the previous anodic sweep, indicating that the localized damage produced in the passive layer is maintained, leading to a greater surface of the steel exposed to the electrolyte, and the material continues corroding in the test solution at a greater rate than during the previous sweep. In this way, a repassivation potential ($E_{rep}$) can be identified for both steels, as indicated in Table 3. Therefore, the oxide layer present on the steel samples cannot self-repair, although the magnitude of the electrical polarization is decreased. Conversely, for the titanium-based alloys in general, and more particularly for Ti6Al7Nb in both test media types, the hysteresis loop is such that significantly smaller currents are measured during the negative-direction sweep compared to the previous anodic scan. Therefore, the corrosion rate associated with the passive regime is smaller during the subsequent cathodic scan than before the passive layer was generated, indicating effective protection of the metal surface.

### 3.2. Scanning Electrochemical Microscopy Measurements

Amperometric operation in SECM consists of plotting the change in the current measured $i_T$ at the biased microelectrode ME that occurs when it is moved close to the surface

of the biomaterial sample immersed in a simulated physiological solution containing an added redox mediator (ferrocene-methanol in this work). In this way, the SECM maps depict the changes in local electrochemical reactivity associated with the structure of the material as a function of the electrical polarization applied to it. For the sake of simplicity, the simpler Ringer's solution was chosen for the characterization of the passive regime of Ti6Al7Nb, and ferrocene-methanol added to the test solution at a 0.5-mM concentration was employed as a redox mediator by polarizing the ME tip at +0.50 V. For comparative purposes, a sample of 304 SS was imaged under the same experimental conditions because it undergoes localized passivity breakdown and pitting corrosion within the potential range of the passive regime of Ti6Al7Nb (as shown by the potentiodynamic curves in Figure 10). The changes associated with passivity breakdown and the nucleation of a corrosion pit for 304 SS in aqueous solution at ambient temperature have already been imaged using SECM [51,52].

SECM maps of the surfaces were recorded at different potentials applied to the samples, in a four-electrode configuration where both the tip of the ME and the samples themselves acted as working electrodes. Scans of the surfaces were conducted at a fixed height (approximately 15 μm), in the $X-Y$ directions of the plane at a speed of 15 μm/s. The tip potential was set at +0.50 V vs. Ag/AgCl/KCl (sat.). First, the SECM map was obtained with the samples left unpolarized, effectively at their spontaneous OCP; then, the potential applied to the studied alloy was varied from −0.25 V to +1.00 V vs. Ag/AgCl/KCl (sat.) in 50-mV intervals. Figures 11 and 12 show selected SECM maps recorded at random locations on the Ti6Al7Nb and 304 SS surfaces immersed in Ringer's solution at 37 °C, respectively. The maps are plotted using the normalized current values, *I*, determined at each location in the measuring grid.

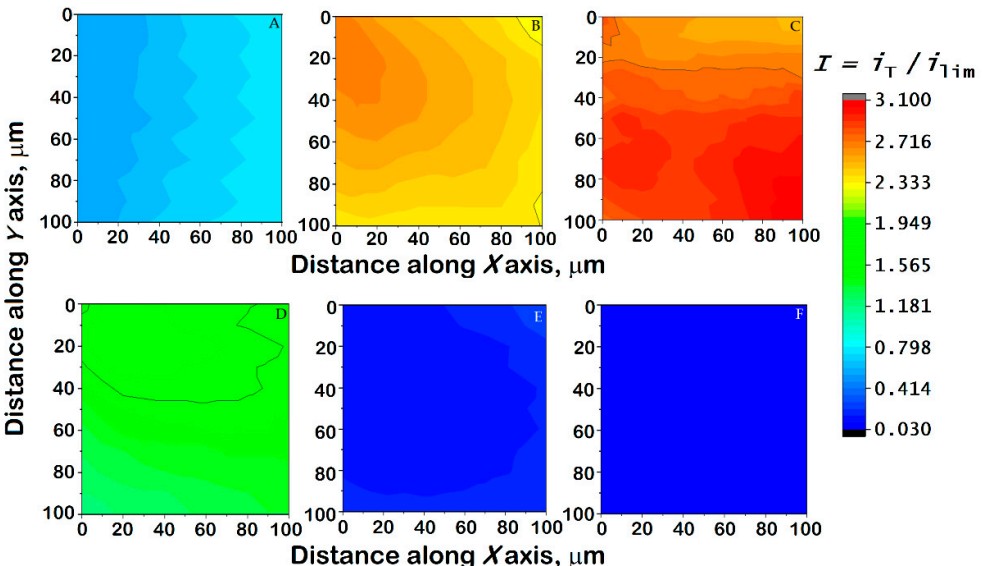

**Figure 11.** Images generated by SECM of a Ti6Al7Nb sample immersed in Ringer's solution + 0.5 mM ferrocene-methanol at 37 °C while (**A**) left unpolarized (i.e., at its corresponding OCP value), and (**B–F**) polarized using the WE#2 connection of the bipotentiostat built into the instrument. Substrate potential: (**B**) −0.20, (**C**) 0, (**D**) +0.20, (**E**) +0.40, and (**F**) +0.60 V vs. Ag/AgCl/KCl (sat.). Tip–substrate distance: 15 μm; scan rate: 15 μm/s; tip potential: +0.50 V vs. Ag/AgCl/KCl (sat.). The images represent a scan of 100 μm × 100 μm along the *X* and *Y* directions.

For both the Ti6Al7Nb alloy and the 304 SS, the current variations mostly reflect the topographic influence when the map was recorded while the materials were left unpolarized at their corresponding OCP value, as evidenced by normalized current values lower than one. This reveals that, despite the spontaneously attained potential being sufficiently cathodic (cf. Table 3) to promote the regeneration of the ferrocene-methanol and the pro-

motion of positive feedback, the passive layer hinders any electrochemical reaction of the redox mediator. Therefore, only a very small tilt of the samples that could not be completely compensated can be observed in Figures 11A and 12A. However, it must be observed that the contribution of this small tip due to sample positioning cannot be observed in any of the remaining maps, and they can therefore be considered to depict local variations in the electrochemical response due to the combination of sample polarization and eventual heterogeneous reactivity.

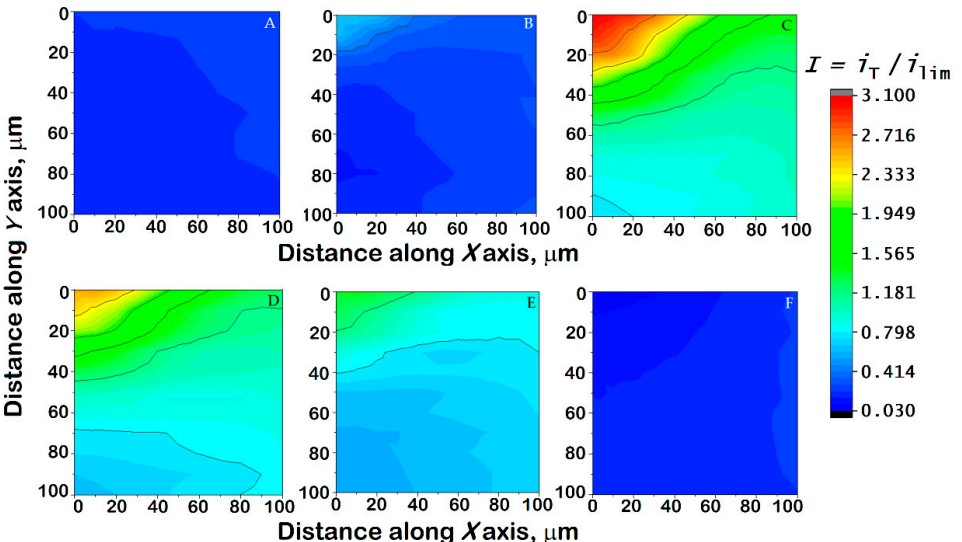

**Figure 12.** Images generated by SECM of a 304 SS sample immersed in Ringer's solution + 0.5 mM ferrocene-methanol at 37 °C while (**A**) left unpolarized (i.e., at its corresponding OCP value), and (**B–F**) polarized using the WE#2 connection of the bipotentiostat built into the instrument. Substrate potential: (**B**) −0.25, (**C**) −0.20, (**D**) 0, (**E**) +0.20, and (**F**) +0.40 V vs. Ag/AgCl/KCl (sat.). Tip–substrate distance: 15 μm; scan rate: 15 μm/s; tip potential: +0.50 V vs. Ag/AgCl/KCl (sat.). The images represent a scan of 100 μm × 100 μm along the X and Y directions.

When a negative potential is applied to the samples, the normalized current measured at the tip becomes greater than unity, indicating the occurrence of positive feedback on both surfaces; this occurs as a consequence of both surfaces becoming more electrochemically active than at the OCP, thus being able to regenerate the redox mediator consumed at the ME tip. This is observed in both samples for the application of substrate potentials of −0.20 and 0.00 V vs. Ag/AgCl/KCl (sat.) (cf. Figures 11B,C and 12C,D). Interestingly, the 304 SS sample still shows a mostly inert surface towards mediator regeneration at the applied potential of –0.25 V vs. Ag/AgCl/KCl (sat.) (Figure 12B), reflecting a thicker oxide layer. Such electrochemical activation is greater and more heterogeneously distributed in steel than in the titanium-based alloy, as reflected by larger current values. In addition, on the SECM map recorded for 304 SS, it is possible to observe localized areas of high activity, suggesting local activation. Likewise, these high values may also indicate the release of $Fe^{2+}$ ions from the surface that are detected by the ME surface at the same potential through a generation-collection mode [51,52], which suggests the initiation of degradative processes on the steel. With the increase in the potential applied to the substrate towards positive values, the currents measured at the tip decrease progressively until values close to those obtained at the OCP are measured again. This feature is observed when the samples are subjected to a potential ≥+0.20 V vs. Ag/AgCl/KCl (sat.), although a certain amount of localized electrochemical activity remains on 304 SS due to the formation of a less stable passive layer and the possible release of iron ions. The application of a substrate potential in excess of +0.25 V vs. Ag/AgCl/KCl (sat.) suppresses the positive feedback effect, as the surfaces do not have a sufficiently cathodic potential to regenerate the mediator. Finally, further polarization at more positive potentials may lead to the establishment of redox

competition for the mediator between the tip and the substrate, and its magnitude depends on the conductivity of the surface layers present on the alloys. This explains the small currents measured at the tip in Figures 11E,F and 12F. Most importantly, no localized domains of enhanced electrochemical activity are detected at any applied potential for the Ti6Al7Nb sample, supporting that the oxide layer responsible for its passive regime is compact and mostly homogeneous on the micrometer scale. Furthermore, it becomes progressively more insulating (i.e., hindering redox mediator regeneration) as it thickens when more positive potentials, within the potential range reached in the human body, are applied. Conversely, competition between oxide layer growth and localized breakdown leading to the nucleation of corrosion pits occurred for 304 SS, indicating that the passive layer is not effective for corrosion protection with increasing anodic polarization.

### 4. Conclusions

All of the alloys studied in this work are capable of forming an oxide layer, although significant differences can be distinguished in their compactness and corrosion resistance on the basis of conventional electrochemical characterization. Since the electrochemical impedance spectra could be satisfactorily fitted using two time constants, the formation of bilayer oxide films with different sealing characteristics was considered. Among them, the titanium-based materials exhibited the most protective oxide layers (higher resistance values) in Ringer's solution at 37 °C, which was employed as a testing media for the comparison of the different materials. Interestingly, although the choice of physiological solution has an impact on the compactness of the oxide film spontaneously formed after immersion of a fresh polished surface, Ti6Al7Nb shows quite similar total resistance values—albeit with different contributions of the inner and outer layers.

The observations drawn from the electrochemical impedance studies are supported by the cyclic polarization curve measurements, despite the perturbation effect of sample polarization. In particular, the Ti6Al7Nb alloy exhibited polarization plots with a negative hysteresis curve, demonstrating that this material can achieve a stable passive behavior within the range of electrical potentials that can be experienced in the human body. Although the oxide film is macroscopically stable, the occurrence of localized breakdown and repair events has also been shown.

The SECM results show that the oxide layer formed on the Ti6Al7Nb alloy is more homogeneous and less conductive than that generated on 304 SS, with the observation of lower electrochemical activities at potential values of $E \geq +0.20$ V vs. Ag/AgCl/KCl (sat.).

**Author Contributions:** Conceptualization, R.M.S.; investigation, A.A.-G.; methodology, J.I. and R.M.S.; writing—original draft preparation, A.A.-G., J.I. and R.M.S.; writing—review and editing, J.I. and R.M.S.; visualization, J.I.; and funding acquisition, R.M.S. All authors have read and agreed to the published version of the manuscript.

**Funding:** This work was supported by Grant PID2021-127445NB-I00 funded by MCIN/AEI/10.13039/501100011033 (Madrid, Spain) and by the European Union Next Generation EU/PRTR (Brussels, Belgium).

**Institutional Review Board Statement:** Not applicable.

**Informed Consent Statement:** Not applicable.

**Data Availability Statement:** The raw/processed data required to reproduce these findings cannot be shared at this time as the data also form part of an ongoing study. They will be available on request.

**Conflicts of Interest:** The authors declare no conflict of interest.

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
