# Peer review of "Corrosion Performance of Ti6Al7Nb Alloy in Simulated Body Fluid for Implant Application Characterized Using Macro- and Microelectrochemical Techniques"

_coatings, doi:10.3390/coatings13061121_

Round 1
Reviewer 1 Report
In this manuscript, authors applied the macro and micoelectrochemical techniques to investigate the corrosion performance of the Ti6Al7Nb alloy. The results indicated that the alloy exhibits high corrosion resistance, making it as a good candidate for biomaterial applications. In general, the manuscript is well organized and the work is interesting. However, there are still some issues to be addressed before its acceptance.
1. One or two sentences are required at the beginning of abstract to present the background or aim of this work.
2. More solid data should be provided in abstract section.
3. More detailed on the raw materials should be provided, such as the component ratio of the used alloy.
4. Three-line table should be applied for better scientific expression.
5. Error bars should be added in some of the figures.
6. Some figures should be modified with a better readability, especially the quite small texts.
7. There are too many too old references, which is better to be deleted or replaced with recent articles to show the novelty of this work.
8. More introduction, discussion and comparison on the corrosion and corrosion mechanism should be provided with some recent supporting articles: J. Eur. Ceram. Soc. 43, 9 (2023): 4114-4123; J. Eur. Ceram. Soc. 43, 2 (2023): 612-620; etc.
9. There are still some typos and grammar issues in the manuscript. Authors should carefully recheck the whole manuscript.
Author Response
- One or two sentences are required at the beginning of abstract to present the background or aim of this work.
Agreed. Following this valuable comment, the beginning of the Abstract section has been expanded with the yellow-highlighted text in the revised manuscript.
- More solid data should be provided in abstract section.
Agreed. The abstract now contains more solid data obtained using both the conventional electrochemical methods and the microelectrochemical methods.
- More detailed on the raw materials should be provided, such as the component ratio of the used alloy.
Done.
- Three-line table should be applied for better scientific expression.
The Tables have been edited for easier readership.
- Error bars should be added in some of the figures.
Agreed. Information on the data variability or error has been included in the revised manuscript, although we considered preferable to be included in the Tables than in the Figures for the sake of readership.
In addition, quality of data analysis can also been deduced by comparing the experimental data (plotted as scattered points) with the simulated data (plotted as solid lines)
- Some figures should be modified with a better readability, especially the quite small texts.
Done. The figures were modified using bigger letter sizes.
- There are too many too old references, which is better to be deleted or replaced with recent articles to show the novelty of this work.
Done. 13 old references have been replaced with the same number of new references.
- More introduction, discussion and comparison on the corrosion and corrosion mechanism should be provided with some recent supporting articles: J. Eur. Ceram. Soc. 43, 9 (2023): 4114-4123; J. Eur. Ceram. Soc. 43, 2 (2023): 612-620; etc.
Agreed. We understand the Reviewer is showing examples of how to improve our paper in that respect, although by using relevant references for those indicated in his report cannot be related to our work since:
(1) J. Eur. Ceram. Soc. 43, 9 (2023): 4114-4123. This paper reports on the thermal corrosion of Yb4Hf3O12 ceramics at 1400 ºC. The work does not involve electrochemical corrosion, neither is a Ti-based material. A completely different mechanism operates in chemical oxidation (which is the case of thermal corrosion) from electrochemical corrosion where the mechanism involves microgalvanic cells.
(2) J. Eur. Ceram. Soc. 43, 2 (2023): 612-620. This paper reports on the influence of water vapor on Yb4Hf3O12 ceramics at 1400 ºC. The same observations that in the previous case.
As result, we have included 3 new references describing the corrosion mechanism of biomaterials in the human body and how to measure it. In addition, we have described the phenomenon in the Introduction Section by inserting a paragraph:
"Corrosion of metallic implant materials in a physiological environment is due to electrochemical processes [9,24,34]. That is, distributed local microcells consisting of anodes (sites where metal oxidation occurs) and cathodes (sites where certain chemical species are reduced) form because the material is in direct contact with an aqueous electrolytic fluid phase. The corrosion rate is affected by local chemical heterogeneities in the material which give rise to a galvanic coupling [10]. Although cathodic and anodic sites may sometimes occur close to each other on the metal surface, the two reactions cannot occur simultaneously at the same location. Since these processes are electrochemical in nature, they can be best studied using electrochemical techniques [23,25,35,37,38]."
- There are still some typos and grammar issues in the manuscript. Authors should carefully recheck the whole manuscript.
The revised manuscript has been revised with an English editor.
Reviewer 2 Report
The paper is devoted to the electrochemical study of the properties of the Ti6Al7Nb alloy with an emphasis on physiological applications. A comparative analysis with well-known materials such as Ti6Al4V and stainless steel has been carried out. According to the authors, spatially resolved electrochemical measurements of Ti6Al7Nb were carried out for the first time, which showed a high uniformity of this material in terms of corrosion resistance. Materials and methods are described clearly enough, the results are presented and discussed with the necessary degree of detail. The paper gives a good impression of scrupulously executed and practically significant work. I think the paper should be published.
The paper has a number of comments:
1. Does the spatial resolution of the method match the size of the electrode (25 µm)? In any case, it is worth specifying the spatial resolution directly.
2. Fig. 3. The nature of the nonmonotonic section of the curve in Ringer's solution (about 250 s) is not entirely clear.
3. The authors used two solutions (Ringer’s and Hank’s). It is desirable to describe in more detail the differences between these solutions from an applied point of view. Which one is closer to the situation in the human body? What physiological situations (different parts of the body) are modeled by these solutions? The range of applications of the studied materials depends on this.
4. Line 446 - the scanning speed is 50 µm / s, Lines 126 and 432 - 15 µm / s. Apparently this is a typo. What is the correct meaning?
Author Response
- Does the spatial resolution of the method match the size of the electrode (25 µm)? In any case, it is worth specifying the spatial resolution directly.
Thank you for noticing. It has been included in the Experimental Section.
- Fig. 3. The nature of the nonmonotonic section of the curve in Ringer's solution (about 250 s) is not entirely clear.
Agreed. The reason for that section has been described in more detail in the revised manuscript as highlighted in yellow.
- The authors used two solutions (Ringer’s and Hank’s). It is desirable to describe in more detail the differences between these solutions from an applied point of view. Which one is closer to the situation in the human body? What physiological situations (different parts of the body) are modeled by these solutions? The range of applications of the studied materials depends on this.
Done. These solutions have been chosen as to match the average mineral composition of the physiological fluid in the human body, but only the second has pH buffering characteristics. Since most of the corrosion resistance data for biomaterials is usually determined in Ringer’s solution, that was the artificial electrolyte employed for the comparison of the different materials. But Hank’s solution was also considered for the study of Ti6Al7Nb to further investigate the eventual effect of phosphate and carbonate as the typical buffering systems in the human body. - Line 446 - the scanning speed is 50 µm / s, Lines 126 and 432 - 15 µm / s. Apparently this is a typo. What is the correct meaning?
Thank you for noticing this typo. The correct value is 15 µm/s
Reviewer 3 Report
In this work, the authors presented an interesting study on the chemical and corrosion resistance of the Ti6Al7Nb biomaterial in a simulated physiological environment were characterized using a multiscale electrochemical approach.
Topics that are covered in the paper.
Materials: Rods of as-cast Ti6Al7Nb were produced and compared with commercial rods of c.p. Ti, Ti6Al4V, stainless steel of biomaterial grade 316L and nonbiomaterial grade 304. Also, two simulated physiological media, namely, Ringer’s and Hank’s solutions were used.
Coating: No coating treatment was performed on the different materials.
Characterization techniques: Electrochemical testing was performed in a three-electrode corrosion cell using PARSTAT (model 2263 potentiostat/galvanostat, PowerSUITE software). Microelectrochemical testing was performed using a SECM instrument built by Sensolytics (Bochum, Germany) containing an Autolab bipotentiostat (Metrohm Autolab, Utrecht, The Netherlands).
Results and Conclusions: All the alloys studied in this work are capable of forming an oxide layer, although significant differences can be distinguished in their compactness and corrosion resistance on the basis of conventional electrochemical characterization. Ti6Al7Nb exhibits high corrosion resistance compatible with a fairly stable passive regime, which allows this alloy to be considered an effective alternative to Ti6Al4V for biomaterial applications.
The overall results are interesting. However, some points must be considered and discussed as follows:
1. The authors provide a comparison between different alloys that are not mentioned in the title, such as CP Ti, Ti6Al4V, 316L ss and 304 ss. The authors must provide these materials in the title. Otherwise, the comparison would have no basis for the study.
2. Can the authors explain what is the purpose of making a comparison between CP Ti, Ti6Al7Nb, Ti6Al4V, 316L ss and 304 ss and what is the application?
3. The results of evolution of the passive layer formed on the biomaterial sample are interesting. However, what is the purpose of compare different simulated solutions with Hank’s and Ringer’s solutions on the formation passive layer of Ti6Al7Nb samples?
4. In general, the study and results are interesting. However, the authors must provide a comprehensive explanation on the use and comparison of different materials and simulated solutions. It is evident that the use of different materials and different simulated solutions leads on different passive layer results. In this sense, what is the purpose of the study using different simulated solutions? why the results of different materials are compared? If the results are considered for biomaterials applications, why is 304 ss considered in this study? 304 ss can probably be omitted in this study.
Author Response
- The authors provide a comparison between different alloys that are not mentioned in the title, such as CP Ti, Ti6Al4V, 316L ss and 304 ss. The authors must provide these materials in the title. Otherwise, the comparison would have no basis for the study.
The title and the abstract have been modified to account for the comparison of the various materials as well.
- Can the authors explain what is the purpose of making a comparison between CP Ti, Ti6Al7Nb, Ti6Al4V, 316L ss and 304 ss and what is the application?
The study is mainly addressed to characterize the biocompatibility of Ti6Al7Nb in terms of corrosion resistance, as a potential replacement for cp Ti (which is mechanically less compatible than the alloys due to a higher Young modulus) and Ti6Al4V (due to the toxicity of V).
On the other hand, the physicochemical models usually employed to describe a passive alloy have been developed for stainless steels (particularly 304 SS in most cases), and they are extended to describe Ti and its alloys. But passivity breakdown leading to localized pitting corrosion is regarded to occur for the stainless steels while regarding Ti-based materials to be stable. But the current knowledge on passivity is that both systems experience passivity breakdown through a similar electrochemical mechanism, although with a great difference in pitting susceptibility. Therefore, the electrochemical characteristics of Ti6Al7Nb have been studied and compared to Ti-based biomaterials and to stainless steels.
- The results of evolution of the passive layer formed on the biomaterial sample are interesting. However, what is the purpose of compare different simulated solutions with Hank’s and Ringer’s solutions on the formation passive layer of Ti6Al7Nb samples?
These solutions have been chosen as to match the average mineral composition of the physiological fluid in the human body, but only the second has pH buffering characteristics. Since most of the corrosion resistance data for biomaterials is usually determined in Ringer’s solution, that was the artificial electrolyte employed for the comparison of the different materials. But Hank’s solution was also considered for the study of Ti6Al7Nb to further investigate the eventual effect of phosphate and carbonate as the typical buffering systems in the human body.
- In general, the study and results are interesting. However, the authors must provide a comprehensive explanation on the use and comparison of different materials and simulated solutions. It is evident that the use of different materials and different simulated solutions leads on different passive layer results. In this sense, what is the purpose of the study using different simulated solutions? why the results of different materials are compared? If the results are considered for biomaterials applications, why is 304 ss considered in this study? 304 ss can probably be omitted in this study.
In addition to the answer given to ítem #2 above, there are no scanning microelectrochemical studies on orthopaedic-quality stainless steels until now in the literature, but we could only find data on 304 SS. The same applies to c.p. Ti and nitinol (a Ni-Ti alloy) but not to Ti6Al4V yet. Therefore, in order to sustain our data in comparison with the literature, the given choice of materials was made. Therefore, we would prefer not removing the data obtained for 304 SS, since we clearly stated that this alloy has not applicability as biomaterial, but for the sake of better describing and supporting the multiscale electrochemical findings raised for Ti6Al7Nb in this work, that are related to the properties of its passive regime (or state).